# Recovery coupling in multilayer networks

Michael M. Danziger ● [1✉] & Albert-László Barabási[1,2,3]

The increased complexity of infrastructure systems has resulted in critical interdependencies between multiple networks—communication systems require electricity, while the normal functioning of the power grid relies on communication systems. These interdependencies have inspired an extensive literature on coupled multilayer networks, assuming a hard interdependence, where a component failure in one network causes failures in the other network, resulting in a cascade of failures across multiple systems. While empirical evidence of such hard failures is limited, the repair and recovery of a network requires resources typically supplied by other networks, resulting in documented interdependencies induced by the recovery process. In this work, we explore recovery coupling, capturing the dependence of the recovery of one system on the instantaneous functional state of another system. If the support networks are not functional, recovery will be slowed. Here we collected data on the recovery time of millions of power grid failures, finding evidence of universal nonlinear behavior in recovery following large perturbations. We develop a theoretical framework to address recovery coupling, predicting quantitative signatures different from the multilayer cascading failures. We then rely on controlled natural experiments to separate the role of recovery coupling from other effects like resource limitations, offering direct evidence of how recovery coupling affects a system's functionality.

[1] Network Science Institute, Northeastern University, Boston, MA, USA. [2] Division of Network Medicine, Department of Medicine, Harvard Medical School, Boston, MA, USA. [3] Department of Network and Data Science, Central European University, Budapest, Hungary. ✉email: mmdanziger@gmail.com

As critical infrastructure systems have increased in size and complexity, so has the interdependence between them—communication systems require electricity from the power grid, whose functioning and maintaining relies, however, on communication systems. Both networks rely on the transportation system for repairs, and in turn, transportation needs both electrical power and a functioning communication system. These multiple interdependencies, and their consequences for resilience, have inspired an extensive literature on coupled multilayer networks, crossing disciplinary boundaries[1–10].

The common hypothesis behind the current multilayer network modeling framework is one of hard interdependence, where a node or link failure in one network causes node or link failures in another network, which in turn may induce additional failures in the original network, resulting in a domino-like cascade of failures across multiple systems[3] (Fig. 1a). Despite the many modeling insights it has offered, evidence of such hard cascading failures remains limited in real systems. For example, while communications and some transit networks do depend directly on electricity, failures in these networks rarely cause electrical failures[8]. Furthermore, while cascading failures in the electric grid are well documented[11–16], despite a decade-long body of literature on the subject, we continue to lack convincing empirical evidence of these cascades, inducing cascades of failures in other infrastructure systems.

While direct evidence of hard coupling across multiple networks is limited, there are multiple accounts of interdependencies not considered by the current modeling frameworks, those induced by the recovery process[8,17–19]. Indeed, the repair and the recovery of a network following a local or global failure requires resources typically supplied by other networks. For example, restoring failed power components requires that the repair crews have access to transportation (road networks) and coordination through communications (cellular networks and internet). If the support networks are not fully functional, the delivery of resources critical for recovery will be slowed or impaired (Fig. 1b). Indeed, while a blocked road or an internet outage in a given location will not cause a power outage, it may delay the repair of power outages in the affected area. And because the damage may continue regardless of the system's ability to recover, impaired recovery could eventually lead to a system's collapse. Such recovery-based interdependencies were well documented in the aftermath of Hurricane Sandy: at least 85 incidents of recovery interdependence were reported, including the dependency of the power grid's recovery on other networks[17].

## Results

Here we show how recovery coupling affects a network's functionality, finding that its signatures and dynamics are different from the much-studied multilayer cascading failures, as well as from interdependent networks with coupling[20–24]. To empirically test the developed framework, we collected data on millions of power grid failures in the contiguous United States, finding evidence of striking nonlinear behavior in recovery following large perturbations, consistent with the model predictions.

**Network damage and recovery at constant rates**. Consider two infrastructure systems $X$ and $Y$, each composed of $N$ elements (nodes). Each network is described by its adjacency matrix, $X_{ij}$ and $Y_{ij}$, and we label the nodes geographically so that co-located nodes $x_i$ and $y_i$, have the same index $i$. At any moment, each node can either be functional ($x_i = 1$, $y_i = 1$) or non-functional ($x_i = 0$, $y_i = 0$). A non-functional node can cause secondary damage either by isolating its neighbors from the rest of the network, or via cascading mechanisms[16]. Though a single node or link failure can render other parts of the network nonfunctional, once the initial failure is repaired, typically the secondary failures will also return to functionality[25]. For example, though a downed power

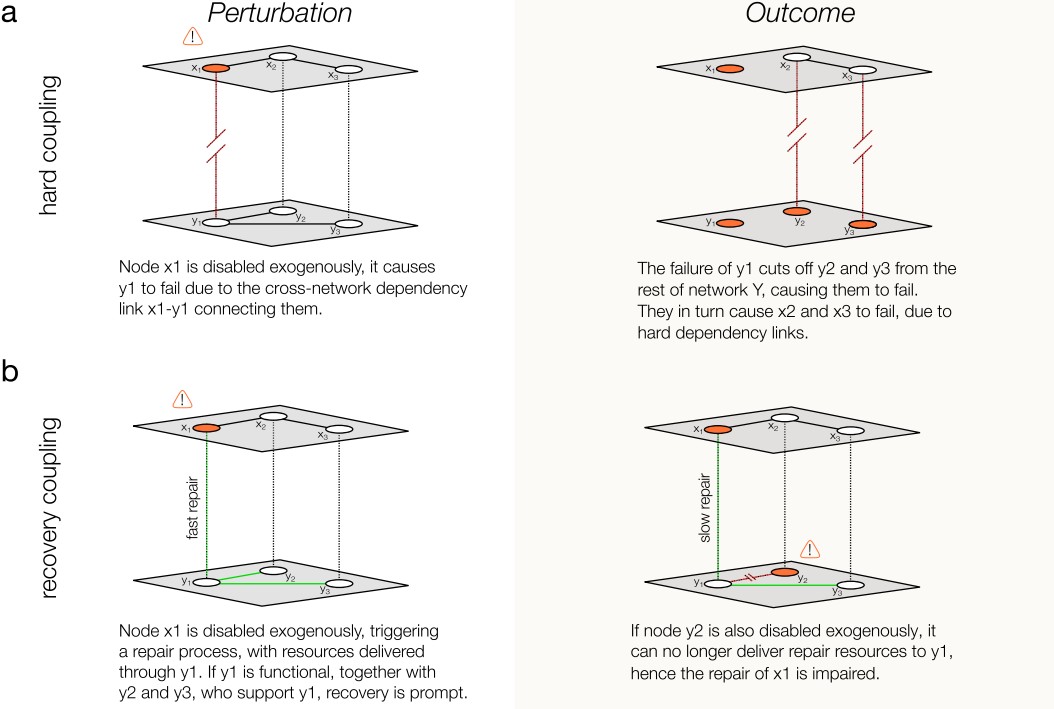

**Fig. 1 Damage and recovery in interdependent networks. a** Under the hard coupling model, when node $x_1$ fails, it causes a cascade across both networks that disables the entire system. **b** With recovery coupling on the same network, when node $x_1$ fails it is repaired using resources from network $Y$ delivered through node $y_1$. Failures in network $Y$ will impair that repair process.

line may cut off power to many homes, once the line is repaired, the power will be restored to each home without needing the individual repair of each component.

Assuming a constant damage rate $\gamma_\mu^d$ and a constant repair rate $\gamma_\mu^r$, the fraction of primary failed nodes in each network $f_\mu$ evolves in time as

$$\dot{f}_\mu = \gamma_\mu^d(1 - f_\mu) - \gamma_\mu^r f_\mu, \tag{1}$$

reaching the equilibrium damage fraction

$$\langle f_\mu \rangle = \frac{1}{1 + \gamma_\mu^r/\gamma_\mu^d}. \tag{2}$$

The damage rate $\gamma_\mu^d$ is largely exogenous and determined by weather, accidents, or component failures. The repair rate $\gamma_\mu^r$, in contrast, is determined by the resources available for repair, such as crew and supplies.

Equations (1)–(2) predict a linear relationship between the number of damaged nodes and the number of repairs executed within a given time window, analogous to the elastic balance between displacement and restoring forces in stress-strain relationships in materials science[26,27]. A constant damage rate $\gamma_\mu^d$ leads to $\gamma_\mu^d N$ sites being damaged at any time, and temporal variability can be modeled by replacing the constant $\gamma_\mu^d$ with a stochastic variable from a representative distribution (see supplementary note 3).

**Observed outage recovery behavior**. To empirically test the validity of elastic recovery, we built an Outage Observatory, a suite of continually running web crawlers that record live-updating outage maps[28–30] from electrical utilities around the United States (Fig. 2a). During 2019 the Observatory recorded over 5 million power outages, capturing the geographic location and time of each outage and the repair time for each incident (Fig. 2e). By comparing the number of repairs and outages occurring in a utility at any time, we can construct the damage-repair curves for each utility (Fig. 2b and c), finding that for most utilities the recovery follows the linear response of Eq. (1) 95% of the time, whose slope provides the repair rate (supplementary note 5 for details). However, we also observed multiple large disruptions, for which the number of repairs systematically and significantly deviates from the linear pattern characterizing the elastic behavior (Fig. 2d). We have been able to link many of these to large events such as severe winds, rainfall, snowfall, and fires. For example, a derecho system that struck the Northern Midwest on 19 July 2019[31] caused over 55,000 outages, resulting in over 60 million lost customer hours. Each perturbation impacts the power grid and its support systems in different ways, hence the precise deviation from linearity cannot be inferred from the number of outages alone. Though each large failure has its unique cause and recovery dynamics, when we place all perturbations on the same graph we observe a remarkable universality, finding that all large events display similar nonlinear behavior (Fig. 2d).

**Modeling recovery coupling**. The loss of elasticity during extreme perturbations indicates that the hypothesis of a constant repair rate is not sufficient to explain the system's behavior. Given that the repair process requires resources from other networks, we hypothesize that a multi-network approach could explain the observed deviation. To model the observed dependency, we allow the repair rate $\gamma_{X,i}^r$ of the primary network $X$ (e.g., power grid) at node $i$ to depend on the state of the support network $Y$ (e.g., road or communication network) at the same location (Fig. 1b),

obtaining

$$\gamma_{X,i}^r(t) = g(\langle y \rangle_i(t)) = g(1) - g'(1)(1 - \langle y \rangle_i(t)) + o((1 - \langle y \rangle_i(t))^2), \tag{3}$$

where $g(x)$ is an unknown function that represents the functional dependence of the repair rate of system $X$ on the state of network $Y$ around site $i$, which we assess with the network average

$$\langle y \rangle_i(t) = \frac{1}{k+1} \sum_i Y_{ij} y_j(t), \tag{4}$$

to capture the fact that repair resources are drawn from the neighborhood of the failure and are affected by the networks which supply them. Thus $\langle y \rangle_i$ may represent the dynamically evolving accessibility, or availability of electricity. In (4), the variable $\langle y \rangle_i(t)$ captures the temporally evolving local state of network $Y$, which may itself co-evolve with the state of the nodes in network $X$ if dependencies exist between the two systems.

Denoting with $\gamma_X^{r,0} = g(1)$ the elastic repair rate and with $\alpha = g'(1)/g(1)$, we obtain

$$\gamma_{X,i}^r(t) = \gamma_X^{r,0}(1 - \alpha(1 - \langle y \rangle_i(t))), \tag{5}$$

enabling us to describe the expected behavior of $g(x)$ to first order with the assumption that $\alpha \in (0, 1)$. Specifically, we assume that damage in $Y$ will not improve repair in $X$ ($g'(1) \geq 0 \rightarrow \alpha \geq 0$) and that the repair rate must remain positive ($|g'(1)| \leq |g(1)| \rightarrow \alpha \leq 1$).

If damage is sporadic and uncorrelated across both systems, the simultaneous failure of $x_i$ and $y_i$ for a given $i$ is rare, and when the failures are limited to a single network, recovery is not impaired (Fig. 1b). However, if damage in $X$ and $Y$ is correlated in time or space, simultaneous damage of nearby sites in $X$ and $Y$ will occur with higher frequency and based on Eq. (5) we expect a reduction in the repair rate. Such correlations are often caused by severe weather events, the main source of disruptions to all infrastructure systems in the United States[32–34]. These events are highly localized in time and space, simultaneously damaging the electric, communications, and transportation networks. Hurricane Sandy, for example, induced failures across the power grid and communications networks (downed lines, flooded control centers) and transportation networks (flooded roads). These simultaneous failures lead to recovery delays, as power outages could not be repaired because roads were flooded. At times, the coupling was bi-directional: some flooded roads had pumping systems for drainage, which could not be operated without electricity[17].

**Recovery coupling case study: Tropical Storm Imelda**. When there are many outages at once, the repair time can also be affected by resource limitations, like a limited number of repair crew members and trucks. Yet resource limitations are expected to impact the whole service area equally. If, however, the slow-down is limited to regions where the support infrastructure is damaged, recovery coupling is the main driving factor. To distinguish between these two mechanisms, we relied on natural experiments, when exogenous shocks simultaneously affected the electrical network and its support networks. In September 2019, Tropical Depression Imelda caused widespread power outages and flooding in Houston, Texas, and the surrounding area (Fig. 3a). We analyzed the duration for all power outages in the vicinity of flooded roads, using areas without flooding as control, allowing us to test whether the slowdown in outage repairs was due to system-wide drains on resources or on the dependence of the repair rate on road networks. We also considered a temporal control, inspecting the repair times of outages reported over the previous 60 days in the same area (Fig. 3b, e). We find that the

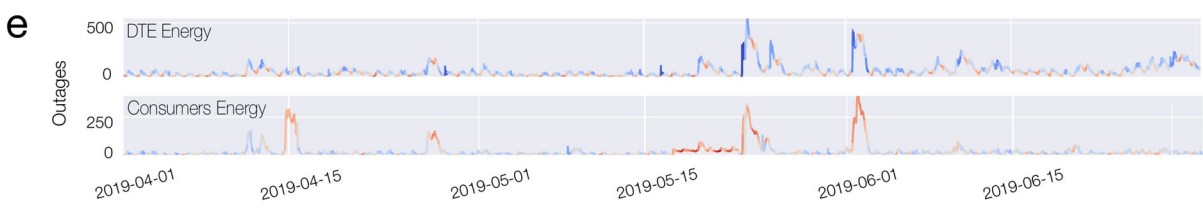

slowdown in outage restoration is heavily localized in both space and time around the flooded roads: while more than 95% of the outages located more than 30 km from the flooded roads were repaired within 10 h, 40% of the failures occurring within 5 km of a flooded road remained unrepaired after 10 h. Furthermore, even during the storm, outages far from flooded roads were repaired at the same rate as without a storm (spatial control, Fig. 3e). The

observed separation of outage survival curves at different distances from flooded roads offers direct evidence of multilayer recovery coupling, illustrating how damage in a non-electrical infrastructure impacts the functionality of the electrical infrastructure.

Further evidence of the proposed phenomenon is provided by the coexistence of elastic behavior far from the flooded roads with

**Fig. 2 Elastic and inelastic recovery in the power grid. a** Locations of outages recorded by the Outage Observatory, colored by the utility serving that area. **b**, **c** Repairs executed vs total outages recorded for each 2-h window for DTE Energy (**b**) and Consumers Energy (**c**), two large utilities in Michigan. An elastic response implies that a constant fraction of outages are repaired at any given time. When the number of outages is small, the response is elastic but when the system experiences a large number of outages it can become increasingly inelastic. Red and blue indicate the deviation from elastic response in the downward and upward direction, respectively. **d** The elastic residual is the difference between the observed repair and the predicted repair based on an elastic response. Comparing the 30 utilities with the most outages, we find a universal downward deviation, (more red points). **e** The number of outages observed at each moment for DTE Energy and Consumers Energy. Because the deviation from elasticity can be quantified for each time window, we can use the color map of panels (**b**, **c**) to indicate system elasticity over time. Measurements of elastic and inelastic recovery for more utilities can be found in Supplementary Figs. 3 and 4.

inelastic behavior near them (Fig. 3c, d). We note that the repair amounts are not only below the elastic prediction, but decrease with increased damage, in line with the prediction that the deviation from elasticity is not due to resource constraints which tend toward saturation of repair per unit time (Supplementary note 2 and ref. [35]).

**Recovery coupling simulations and phase space**. To understand the implications of recovery coupling for multilayer network resilience, we consider the symmetric case in which the network structure, damage, and recovery parameters are the same in both systems. Since the two systems support each other, we let the repair rate of $Y$ be influenced by the state of $X$ in the same manner as Eq. (5): $\gamma_{Y,i}^r(t) = g(\langle x \rangle_i(t))$. In the symmetric case $f_x = f_y = f$, leading to a single equation that governs the state of the system. If the failures are uniformly distributed, we can use percolation theory[36,37] to analytically derive the equation that governs the expected fraction of primary failures in the coupled system,

$$f = \frac{1}{1 + \frac{\gamma^{r,0}}{\gamma^d}(1 - \alpha(1 - u(1 - f)))},$$ (6)

where $u(x)$ is the probability that a link does not lead to the largest connected component when a random fraction $1 - x$ of the nodes are removed, and is determined by the network topology. Equation (6) has one or two stable solutions depending on the value of the control parameter $\frac{\gamma^{r,0}}{\gamma^d}$. The non-symmetric case has similar results, as we shown in supplementary note 1 and Supplementary Fig. 1. In contrast, the uncoupled case (2), which we recover from (6) for $\alpha = 0$, has a single stable solution. The new solution describes a stable fixed point at $f = 1$ (all nodes failed), which persists even for high recovery rates $\frac{\gamma^{r,0}}{\gamma^d}$ (see Fig. 4a). The existence of two stable solutions for $f$ for the same recovery rate $\frac{\gamma^{r,0}}{\gamma^d}$ indicates that for a wide range of conditions, recovery coupled networks are resilient: they display functionality comparable to the uncoupled case and return to full functionality following small perturbations[38,39]. However, a sufficiently large perturbation can force the system to cross the unstable branch, pushing it into a dynamically stable non-functional state (Fig. 4a). This is more likely with correlated perturbations across layers, as we show in Supplementary Fig. 2. The existence of this behavior analytically predicts a "catch 22" phase that follows a sufficiently large disaster: infrastructure system $X$ cannot be repaired because it requires resources from $Y$, and $Y$ cannot be repaired because it requires resources from $X$. The fact that the collapsed state persists even for high repair rates and low damage rates predicts that it is harder to bootstrap a broken system than it is to maintain the functionality of one that is damaged but still working. Synthesizing elastic residual curves (Fig. 4c) like the observations in Fig. 2d, we find that the full coupling $\alpha = 1$ reproduces the shape

of the curve, while lower values of $\alpha$ do not, providing further evidence that the general deviation from elasticity is consistent with recovery coupling.

## Discussion

The 27 September 2003 blackout in Italy is often used to illustrate how the interdependence of communications and electrical infrastructure can cause cascading failures[3,40]. However, a closer look at the sequence of events indicates that though transmission network overload cascades triggered the power outage[41], dependence of repair activities on the communication network which was itself disabled, prolonged the recovery process[42]. Here we demonstrated that such recovery coupling can lead by itself to a collapse of functionality. More importantly, we have shown that the signatures of recovery coupling are directly observable during severe weather events, indicating that the proposed mechanisms have direct relevance to real multilayer networks. Domino-like dependencies, which could co-occur, further amplify this danger, though some interdependencies have been shown to reduce cascading[43].

The data-driven approach presented here enables a more precise understanding of infrastructure interdependence. For example, we find that while the set of flooded roads as a whole caused slowdowns in power outage repairs, some impaired roads had much stronger effects than others. The roads in downtown Houston caused only minor delays when flooded, while in Beaumont and Northeast Houston flooded roads caused severe delays (Fig. 3a). Improving the precision with which we measure infrastructure vulnerability is particularly important in light of aging infrastructure and climate change.

Our findings reinforce the importance of engineering for resilience not only through strengthening critical infrastructure, but also focusing on the socio-technological layers needed to restore the infrastructure when damaged. These recovering systems may supply electricity, fuel or access, or they may provide human connections through social networks, which have also been shown to play a powerful role in disaster recovery[44].

Recovery coupling has relevance for other systems affected by multiple networks. A pertinent example is the impact of loss of healing ability during aging. Living organisms display a fundamental asymmetry between damage and repair, similar to what we observe in infrastructure networks: damage is typically caused by external factors (oxidants, pathogens, shocks, etc.) while repair is endogenous and is governed by multiple coupled networks (regulatory, metabolic and signaling) requiring diverse resources (nutrients, oxygen, immune cells, etc.). From this perspective our work complements recent network-based modeling of the relation between repair and aging[45,46], illustrating how the well-documented loss of healing ability in individual systems[47], can lead to systemic frailty, where the organisim can lose its ability to respond to shocks that it could tolerate under normal conditions[48].

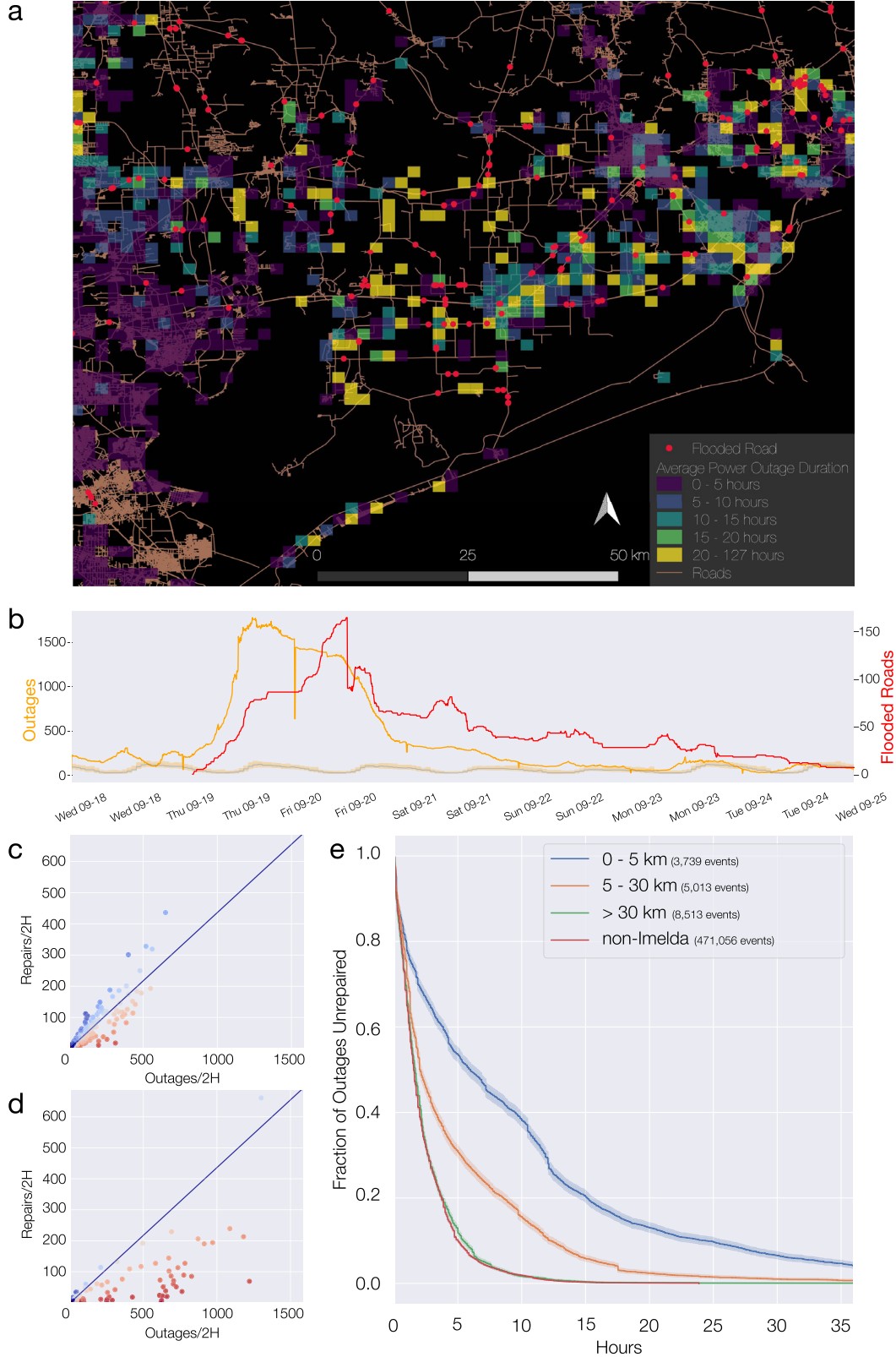

## Methods

**Data**. Outage data were collected by taking regular snapshots (several per hour) of the outage maps published by electric utilities around the United States. Each snapshot contains a geotagged list of all outages active at that time, including transmission and distribution outages. A single outage is reported for each incident, even if many customers are affected. By comparing snapshots from moment to moment, and noting the first and last time that the outage appeared, we can identify the outage's location and duration. For more detail about the data collection, see supplementary note 4 and Supplementary Table 1. To download the data used in this analysis, visit https://github.com/mmdanziger/recovery-coupling.

**Simulation**. We performed discrete-time simulations where at each time point we scan all nodes in each node. For every operational node, we switch it to inopera-

**Fig. 3 Empirical evidence for recovery coupling. a** Average outage duration and location of flooded roads during Tropical Depression Imelda. Delayed restoration occurred primarily around Beaumont and Northeast Houston, where most flooded roads were located. **b** Number of outages (orange) and flooded roads (blue) during Imelda. The shaded orange curve shows the middle quartiles of outages for the same times (hour and day) with no storm, offering a time control. **c** The recovery of outages during Imelda that were far (≥10 km) from flooded roads is well approximated by an elastic (linear) response. **d** Outages near (<10 km) flooded roads show substantial deviation from elastic behavior. The coloring encodes deviation from elasticity as in Fig. 2. **e** Fraction of unrepaired outages grouped according to their distance to the nearest flooded road. Using the Kaplan-Meier estimator, we find statistically significant longer outage durations for outages closer to flooded roads. The spatial control of outages far from the storm and the temporal control of outages from the storm area in the 60 days before the storm, are comparable. In this case, we have taken proximity to damage as a proxy for the network effects of a road closure.

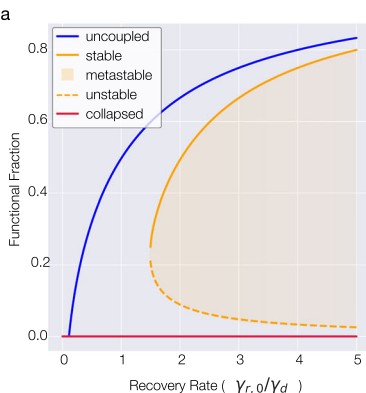
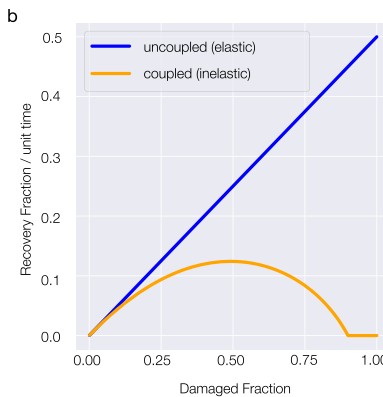
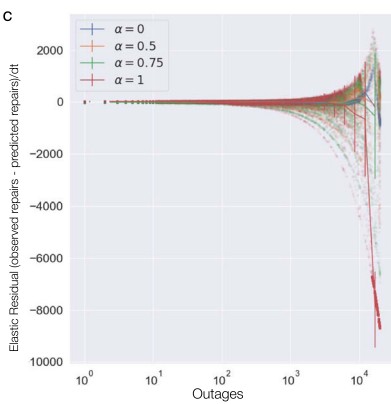

**Fig. 4 Recovery coupling in multilayer networks. a** Comparison between the functionality of uncoupled networks and recovery coupled networks. The uncoupled case (blue line) has a single solution for any repair to damage ratio $\gamma^{r,0}/\gamma^d$, implying that it can recover its functionality after an arbitrarily large perturbation. With recovery coupling the system can function at levels similar to the coupled case (orange line) but the non-functional collapsed state persists as an attractor (red line), implying that for sufficiently large damage, the system can reach a permanently collapsed state. **b** If we inspect the recovery per unit time as a function of concurrent damage amount, we observe a behavior similar to elasticity in materials science. Recovery coupling leads to a sublinear or inelastic behavior, predicting the loss of resilience under heavy damage. **c** The elastic residual plot from bidirectionally coupled random networks shows the same pattern as observed for the real data in Fig. 2d.

tional with probability $\sim \gamma^d$ and for each inoperational node, we switch it to operational with probability $\sim \gamma^r$. The value of $\gamma^r$ is calculated as

$$\gamma^r_{X,i}(t) = \gamma^{r,0}(1 - \alpha(1 - \langle y_i(t-1) \rangle))$$

where $\langle y_i(t-1) \rangle$ is defined as #(operational nodes among $y_i$ and its neighbors at previous iteration) / #(neighbors of $y_i + 1$). In Fig. 4c, for every value of $\gamma^{r,0}/\gamma^d$, we simulate the networks until they converge to a fixed point. We simulate transient behavior in the same manner as shown in Supp. Fig. 2.

## Data availability

The datasets generated during and/or analyzed during the current study are available on GitHub at https://github.com/mmdanziger/recovery-coupling.

## Code availability

The code used for this manuscript is freely available at https://github.com/mmdanziger/recovery-coupling.

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

## Acknowledgements

A.L.B. was supported by NSF CRISP (1735505) and by ONR N00014-18-9-001. M.M.D. would like to thank D. Aldrich, B. Barzel, S.P. Cornelius, A. Gates, A. Grishchenko, S. Havlin, G. Menichetti R.Q. Wang and H. Wu for the many helpful conversations which advanced this research.

## Author contributions

M.M.D. collected the data, developed the model and analyzed the results. M.M.D. and A.L.B. designed the research and wrote the paper.

## Competing interests

A.-L.B. is the founder of Scipher Medicine, Inc., which applies network medicine to biomarker development, of Foodome, Inc., which applies data science to health, and Datapolis, Inc., which focuses on human mobility. M.M.D. declares no competing interests.
