## [Peer Review File · Nature Communications]

REVIEWER COMMENTS

Reviewer #1 (Remarks to the Author):

The manuscript "Recovering Coupling in Multilayer Networks" present a dynamical system model for capturing the dynamics of failures and recoveries in multilayer networks. The model, is motivated and justified by a large scale study of real disruptions in infrastructures and power-grid networks.

The investigation of the robustness of multilayer networks has been studied extensively in the last years. The modelling framework adopted so far by most of the works considers multiplex interdependencies (Nature, 464(7291), pp.1025-1028) or redundant interdependencies (Physical Review X, 7(1), p.011013). However, these approaches are exclusively based on percolation theory and do not capture the dynamics due to the recovering of nodes in time.

The model proposed in this manuscript is rather new and combines dynamical systems with percolation theory to explain not just mere interdependencies but the role that each network has in guaranteeing the recovering of the other coupled networks.

This model, here developed for the case of a duplex network, determines a phase diagram having a bifurcation. In particular, as a function of the model parameters two phases are distinguished: a phase of fast recovery, where disruptions are easily resolved, and a phase of slow recovering where actually the collective stationary state of the system can be either in a functional phase where the vast majority of the nodes are functional or in a dismantled phase where more of the nodes are damaged.

The work is masterly done: it is highly innovative in the modelling framework and in the way dynamical systems equations are combined to percolation theory. This aspect of the work combined with the notable support of the real data I think will be key for exestablishing the high impact of this work in the field of multilayer networks.

Before suggesting publication in Nature Communications I would however like the authors to address the following points.

1. The phase diagram of the dynamics is obtained for a symmetric duplex network in which the role of network A in speeding up the recovery for network B is assumed to be mirrored by the role of network B in speeding up the recovering of network A, leading to the assumption $f_x=f_y$. If possible I would like to see (maybe in the Supplementary Material) a critical discussion related to this hypothesis. What does this hypothesis entails? Can this hypothesis be broken? How robust is the results if this hypothesis does not hold exactly?
2. The paper concludes with a very interesting observation that this work could be useful to understand aging in biological systems. This observation appears also in the abstract. However in the present narrative of the manuscript this observation comes a bit as an unexpected twist that is not supported by data (maybe the plan for a future paper?) I suggest the authors of either expand on this or reduce the emphasis given to this parallelism closing the paper with something more relevant to the main subject of this work.

Reviewer #2 (Remarks to the Author):

My assessment of the manuscript is highly ambivalent. On the one hand, I think that the outage inventory is of extraordinary value for the study of power system reliability (I assume that the data set is published along with the manuscript, isn't it?) and that a thorough statistical analysis can yield important insights. On the other hand, I find the presentation and analysis of the results highly problematic, as I will explicate in my major comments below. In summary, the formulation via

interdependent networks is unnecessarily complicated in view of the empirical results and the theoretical work is only loosely connected to the empirical work. Due to this ambivalent assessment I come to the conclusion that the manuscript is not suitable for publication in Nature Communication in its current form. I would recommend publication if (a) the presentation and analysis is thoroughly revised and (b) the underlying data is made available to the scientific community.

Major issues:

(A) In my opinion the whole framework of interdependent networks is largely unnecessary to analyze and understand the presented empirical results. In the theoretical explanation in the main manuscript (Equations 3 and 5) the network X does not enter at all, while the network Y enters only via the quantity $\langle y \rangle$. One could now replace $\langle y \rangle$ simply by a spatial parameter which characterizes the accessibility for repair and get exactly the same result.

Hence, the framework of interdependent networks is unnecessarily complicated in view of the empirical results. The authors write "we hypothesize that a multi-network approach could explain the observed deviation". While this is certainly true, this is not the simplest approach and thus not the most appropriate one according to Occam's razor. Actually, I get the impression that the idea of interdependent networks was first, and the empirical results are forced into this theoretical framework. But science should always start from the observations and then develop a theory.

(B) One could actually go one step further and ask whether the quantity $\langle y \rangle$ or the accessibility is needed to explain the observed results. The simplest possible explanation for a non-linear or non-elastic relation between repairs and outage is an depletion of resources as the authors briefly discuss on page 5 and 6.

In figure 3, the authors provide compelling evidence that resource depletion cannot account for the slowing down of repairs after Imelda, an exceptional event after all. But I am not yet convince that this cause can be excluded in general. Let's look at the reasoning in more detail.

- On page 5 the authors write that "resource areas are expected to impact the whole service area equally." I believe that this is true to a large extend, but is there any spatial analysis of repair rates other than that during Imelda in figure 3? As I said before, I am perfectly convinced for the special case of Imelda, but is it possible to generalize from one single exceptional event? Or is there any other spatial data?

- In the SI, the authors write "This indicates that, though limited resources will indeed slow down the recovery, it cannot cause a collapse the way recovery coupling can." I fully agree, but we have no empirical case of a full system collapse in the actual data.

- Furthermore, the authors write "under limited repair resources the amount of repair observed per unit time will rise less quickly than in the fully elastic case but will not ever decrease" which I also agree too. But can the authors actually pinpoint such a decrease with the necessary statistical significance? If yes, I think this reasoning should go into the main text. If no, it is much harder, if not impossible, to exclude resource depletion as a main mechanism (except for the single example of Imelda, of course).

- The authors provide some analytic results in Equation 4 in the SI. However, this relies on the linearization in Equation (1) which is questionable. I would assume that the relation of the repair rate γ^r and the and the number of repairs $\gamma^r * f$ is highly nonlinear. In fact I would expect a threshold behavior: First, the repair rate should be almost constant as enough resources are present. As soon the number of outages exceeds the threshold where all resources are in use, the number of repairs $\gamma^r * f$ should saturate such that the repair rate γ^r would actually decay as $1/f$. How would this model fit to the data? Can we exclude it with the necessary statistical significance?

(C) More generally, I think that the framework of interdependent networks is overrated, if not misleading, in the context of power grid stability and reliability. The highly cited reference [3] has established interdependent networks using the example of the 2003 Italian blackout. From an engineering viewpoint, this is highly problematic. Communication networks did not play a role in this cascade, while elementary aspects of power engineering did, but were not included in the analysis. The authors of the current paper discuss this fact on page 7, which is good.

Still, the authors proceed in a similar way as in reference [3]. The first part of the manuscript is about empirical data and the second part is about a highly abstract model of interdependent networks. In my opinion, both parts are only very weakly connected. As argued above the interdependent network approach is not needed to analyze the data, in the simplified version analyzed in Fig. 4 it is certainly not adequate to describe power system reliability. From an engineering perspective, the figure 4 and the associated discussion weaken the paper.

(D) Finally, one should be aware that the majority of power outages (the smaller ones) does not involve any cascading mechanism. Most outages result from equipment failures in distribution grids or feeders at low or middle voltage grids, which are not or only weakly meshed. Then the loss of a single line can directly disconnects some consumers. An analysis of such outages is often possible without a detailed network approach.

On the contrary, high voltage transmission grids are meshed such that the loss of a single line should not lead to any disconnected elements if the grid is operated properly (this is called the N-1 criterion). However, in some cases we nevertheless see cascading failures leading to large scale outages disconnecting thousand or millions of consumers. Understanding these cascades does require a detailed network approach including the physics of power flows.

Minor issues:

(1) How is „recovery coupling“ defined exactly? The term is used a lot, but a precise definition is lacking. Figure 1 gives an idea (which is nice as such), but not a definition.

(2) Figure 2 gives a compelling evidence that power grid recovery depends on accessibility via road transports. But is this an evidence for “recovery coupling”? The paper and in particular figure 1 imply that recovery coupling is only defined for network but the network aspects are not really present in figure 3. This is not a criticism of the figure as such (which I find great), I just think that the caption is way to unspecific.

(3) As mentioned in the beginning, an open outage repository would be of extraordinary value for power system reliability science. However, the description in the main manuscript is not precise enough. For instance, what exactly do you mean if you count an “outage”? Is it one event where a certain number of customers is disconnected? Or does one outage correspond to exactly one failing element of the grid?

(4) Extending on my previous comment: A methods section in the main manuscript would clearly be very helpful to understand what is done exactly.

(5) Page 4. The authors write “all large events collapse on a single nonlinear curve.” I am not fully convinced by this statement. There is still quite a large degree of variability in the data. Furthermore, when I look into the curves in the Supplementary Figures 2 and 3, I see quite strong differences.

(6) On page 7, the authors summarize their results and give an outlook in the paragraph starting with “The ability to identify”. In my opinion, the conclusions are a little exaggerated. Does this single paper

"open the door" to the large field of infrastructure vulnerability? I recommend to tone down the voice here.

(7) Finally I am a bit surprised that a study on power system reliability contains so few references to papers in power engineering journals. For instance, there are ongoing discussions about self-healing smart power grids – a topic which could be of relevance here.

A possible starting point in this direction can be: Amin, Massoud. "Challenges in reliability, security, efficiency, and resilience of energy infrastructure: Toward smart self-healing electric power grid." 2008 IEEE Power and energy society general meeting-conversion and delivery of electrical energy in the 21st century. IEEE, 2008.

A discussion of the role of storms on distribution grid reliability may be found here: Brown, R. E., et al. "Distribution system reliability assessment: momentary interruptions and storms." IEEE Transactions on Power Delivery 12.4 (1997): 1569-1575.

Some statistical studies of distribution grids reliability indices may be found here: Balijepalli, Nagaraj, Subrahmanyam S. Venkata, and Richard D. Christie. "Modeling and analysis of distribution reliability indices." IEEE Transactions on Power Delivery 19.4 (2004): 1950-1955.

Finally, I strongly recommend to read this great review article on cascading failures and power system blackouts:

Pourbeik, Pouyan, Prabha S. Kundur, and Carson W. Taylor. "The anatomy of a power grid blackout-Root causes and dynamics of recent major blackouts." IEEE Power and Energy Magazine 4.5 (2006): 22-29.

Reviewer #3 (Remarks to the Author):

The paper introduces a theoretical framework based on nonlinear systems and multi-layer networks, with an application on power networks.

The paper is clearly written and presents a convincing framework elegant in its simplicity, that is able to capture the main aspects related to the interdependence of power infrastructures in relation to telecommunications and road networks.

The database considered in the study is consistent and covers a wide temporal and geographical frame.

Results are presented in a clear and convincing way, and in particular the existence of two stable solutions is of paramount importance for both utilities and decision makers that aim to develop tools for the analysis and prediction of catastrophic events, even those not directly related to storms or other atmospheric disasters, as, by example, climate-related issues like heat waves and increased systems failure due to temperature increase or decrease.

In my opinion the paper is certainly worth to be published after the following minor points:

1) Please provide a better description about the events that you did consider for the development of the model. Did the authors only consider big failure events or they also included in the analysis other events that can be easily fixed by automatic procedures? (e.g. disconnection of overloaded stations, failure of TLC systems etc).

2) Please include in the conclusions implications for industry, utilities and policy/decision makers.

Recovery Coupling in Multilayer Networks

Point-by-point response to reviewer comments

Reviewer #1 (Remarks to the Author):

The manuscript “Recovering Coupling in Multilayer Networks” present a dynamical system model for capturing the dynamics of failures and recoveries in multilayer networks. The model, is motivated and justified by a large scale study of real disruptions in infrastructures and power-grid networks.

The investigation of the robustness of multilayer networks has been studied extensively in the last years. The modelling framework adopted so far by most of the works considers multiplex interdependencies (Nature, 464(7291), pp.1025-1028) or redundant interdependencies (Physical Review X, 7(1), p.011013). However, these approaches are exclusively based on percolation theory and do not capture the dynamics due to the recovering of nodes in time.

The model proposed in this manuscript is rather new and combines dynamical systems with percolation theory to explain not just mere interdependencies but the role that each network has in guaranteeing the recovering of the other coupled networks.

This model, here developed for the case of a duplex network, determines a phase diagram having a bifurcation. In particular, as a function of the model parameters two phases are distinguished: a phase of fast recovery, where disruptions are easily resolved, and a phase of slow recovering where actually the collective stationary state of the system can be either in a functional phase where the vast majority of the nodes are functional or in a dismantled phase where more of the nodes are damaged.

The work is masterly done: it is highly innovative in the modelling framework and in the way dynamical systems equations are combined to percolation theory. This aspect of the work

combined with the notable support of the real data I think will be key for establishing the high impact of this work in the field of multilayer networks.

We wish to thank the Reviewer for the positive assessment of our results and the many constructive recommendations that we address below.

Before suggesting publication in Nature Communications I would however like the authors to address the following points.

1. The phase diagram of the dynamics is obtained for a symmetric duplex network in which the role of network A in speeding up the recovery for network B is assumed to be mirrored by the role of network B in speeding up the recovering of network A, leading to the assumption $f_x=f_y$. If possible I would like to see (maybe in the Supplementary Material) a critical discussion related to this hypothesis. What does this hypothesis entails? Can this hypothesis be broken? How robust is the results if this hypothesis does not hold exactly?

We would like to thank the referee for this suggestion. We fully agree that the assumption that the two networks are generated by the same probabilistic process is not expected to hold for many systems. As we show in the equations in the revised Supp. Sec. 1 and the new Supp. Fig. 1, this assumption is not necessary to observe the phases shown in Fig. 3. We have also added a mention of this to the main text.

2. The paper concludes with a very interesting observation that this work could be useful to understand aging in biological systems. This observation appears also in the abstract. However in the present narrative of the manuscript this observation comes a bit as an unexpected twist that is not supported by data (maybe the plan for a future paper?) I suggest the authors of either expand on this or reduce the emphasis given to this parallelism closing the paper with something more relevant to the main subject of this work.

Aging is often described in the professional medical literature terms of "frailty". Among the several clinical definitions of frailty, a feature common to many formulations is a reduced capacity to respond to

perturbations of many sorts. This has been hypothesized to be a driving factor of the aging phenotype and ultimately death. Previous work by Taneja et al. has pointed to the possibility that the phenomenon of frailty may be due to cumulated damage leading to loss of repairability, and has used network-based capacity models to understand these processes. Our work illustrates how this phenomenon can emerge as an interaction between recovery coupled networks. We mention this in the article to show that recovery coupling could have applications beyond infrastructure systems, and could be useful for understanding aging as more data becomes available.

Nevertheless, we agree with the reviewer that to give justice to this subject, we need either a more extensive discussion or a more limited set of claims. In light of the reviewer's comment, we have removed aging from the abstract, and expanded it in the discussion, offering more detail. We wish to thank the reviewer for helping us improve the clarity and scope of the discussion.

Reviewer #2 (Remarks to the Author):

My assessment of the manuscript is highly ambivalent. On the one hand, I think that the outage inventory is of extraordinary value for the study of power system reliability (I assume that the data set is published along with the manuscript, isn't it?) and that a thorough statistical analysis can yield important insights. On the other hand, I find the presentation and analysis of the results highly problematic, as I will explicate in my major comments below. In summary, the formulation via interdependent networks is unnecessarily complicated in view of the empirical results and the theoretical work is only loosely connected to the empirical work. Due to this ambivalent assessment I come to the conclusion that the manuscript is not suitable for publication in Nature Communication in its current form. I would recommend publication if (a) the presentation and analysis is thoroughly revised and (b) the underlying data is made available to the scientific community.

We thank the reviewer for the time and attention devoted to our work. We have followed the reviewers suggestion by revising the manuscript and we wish to confirm that with the paper's publication, *we will be releasing the collected data, making it available to the community*. The data and scripts needed to recreate all of the reported results is available at <https://github.com/mmdanziger/recovery-coupling> We agree with the reviewer that this will serve as a useful resource for future work potentially including alternative analyses and explanations. Next we address the detailed comments, offering a point-by-point response below and a list of changes.

Major issues:

(A) In my opinion the whole framework of interdependent networks is largely unnecessary to analyze and understand the presented empirical results. In the theoretical explanation in the main manuscript (Equations 3 and 5) the network X does not enter at all, while the network Y enters only

via the quantity $\langle y \rangle$. One could now replace $\langle y \rangle$ simply by a spatial parameter which characterizes the accessibility for repair and get exactly the same result.

Hence, the framework of interdependent networks is unnecessarily complicated in view of the empirical results. The authors write “we hypothesize that a multi-network approach could explain the observed deviation”. While this is certainly true, this is not the simplest approach and thus not the most appropriate one according to Occam’s razor. Actually, I get the impression that the idea of interdependent networks was first, and the empirical results are forced into this theoretical framework. But science should always start from the observations and then develop a theory.

We wish to apologize for our notation, that apparently has abstracted the coupled nature of the two systems. Indeed, we would like to clarify that the $\langle y \rangle$ term that appears in equations 3 and 5 is not a static parameter capturing the state of system Y, but rather a temporal variable representing the *local network state* that co-evolves with the state of the nodes in network X. If we are dealing with a transportation network, it does indeed describe a kind of accessibility, but this accessibility term evolves in time according to the state of the transportation network. Considering how transportation-specific network dynamics would affect the resilience of the coupled network system is an excellent idea for future research, but is beyond the scope of this study.

We wish to thank the Referee for bringing this potential misunderstanding to our attention, and we now explicitly indicate the time dependence of y by adding time as a variable in the notation $y(t)$. Following the referee's advice, we have also added a line explaining that this could represent a dynamically evolving accessibility measure in some cases.

We fully agree that a model should be as simple as possible and not "unnecessarily complicated." We find, however, that anything simpler would fail to capture the basic fact of recovery coupling. The fact that the recovery of one form of infrastructure is dependent on the functionality of another form of infrastructure has been widely reported in the disaster and resilience literature, predating our work by decades [Rinaldi, *IEEE control systems magazine* **21** (2001), Ouyang, *Reliability engineering & System safety* **121** 43 (2014)].

What we are proposing with this model is a framework to enable the mathematical analysis of how this coupled dynamics can play out in a networked system.

We followed the referee's suggestion that "**science should always start from the observations and then develop a theory**" by compiling a unique data set, and inspired by the patterns observed in the data we built a model that reflects a well-known but never-before modeled feature of network resilience, coupling of recovery patterns. Finally, we have shown that the model and data are in agreement, up to the resolution provided by the data. There are certainly other possible causes, and we explicitly discuss them in the paper. In previous work, even compiling the data on this scale has not been attempted.

(B) One could actually go one step further and ask whether the quantity $\langle y \rangle$ or the accessibility is needed to explain the observed results. The simplest possible explanation for a non-linear or non-elastic relation between repairs and outage is an depletion of resources as the authors briefly discuss on page 5 and 6.

In figure 3, the authors provide compelling evidence that resource depletion cannot account for the slowing down of repairs after Imelda, an exceptional event after all. But I am not yet convince that this cause can be excluded in general. Let's look at the reasoning in more detail.

- On page 5 the authors write that "resource areas are expected to impact the whole service area equally." I believe that this is true to a large extend, but is there any spatial analysis of repair rates other than that during Imelda in figure 3? As I said before, I am perfectly convinced for the special case of Imelda, but is it possible to generalize from one single exceptional event? Or is there any other spatial data?

- In the SI, the authors write "This indicates that, though limited resources will indeed slow down the recovery, it cannot cause a collapse the way recovery coupling can." I fully agree, but we have no empirical case of a full system collapse in the actual data.

- Furthermore, the authors write “under limited repair resources the amount of repair observed per unit time will rise less quickly than in the fully elastic case but will not ever decrease” which I also agree too. But can the authors actually pinpoint such a decrease with the necessary statistical significance? If yes, I think this reasoning should go into the main text. If no, it is much harder, if not impossible, to exclude resource depletion as a main mechanism (except for the single example of Imelda, of course).

- The authors provide some analytic results in Equation 4 in the SI. However, this relies on the linearization in Equation (1) which is questionable. I would assume that the relation of the repair rate γ^r and the and the number of repairs $\gamma^r \cdot f$ is highly nonlinear. In fact I would expect a threshold behavior: First, the repair rate should be almost constant as enough resources are present. As soon the number of outages exceeds the threshold where all resources are in use, the number of repairs $\gamma^r \cdot f$ should saturate such that the repair rate γ^r would actually decay as $1/f$. How would this model fit to the data? Can we exclude it with the necessary statistical significance?

We fully agree with the referee: It is key to understand if and how resource constraints cause outage delays. The referee is entirely correct that resource constraints are a realistic and presumably ubiquitous feature of power grid repair dynamics. This is precisely why we devoted so much attention to the problem, as highlighted by the referee. In the supplementary information, we even develop a basic resource constraint model, demonstrating that with low resolution measurements the resource limitation and network coupling will have very similar signatures.

And yet, we also know that recovery coupling does occur. We have evidence for this from previous empirical literature, and we have offered direct evidence ourselves in the case of Tropical Storm Imelda. That is why we refrain from claiming that the entire non-linear behavior is driven by one mechanism or another.

We are gratified that the reviewer found our analysis of Imelda to be compelling. Indeed, we feel that this case study is one of the highlights of this work and offers a direct empirical analysis of real-world

interdependence that is unprecedented in its precision. Unfortunately, it is very difficult to locate data streams for other events of this magnitude, where multiple relevant layers are simultaneously accessible.

The significance of a model that predicts a "catch 22"-like collapse is a key point which we feel we may have not explained properly in the manuscript. The reviewer states that "we have no empirical case of a full system collapse in the actual data". This is undoubtedly true. During 2019 in the United States there was not a single instance of total infrastructure collapse. But we know that total collapse was observed during large disasters like the fallout from the earthquakes in Haiti in 2010 and in Fukushima in 2011. By their very nature, disasters of this magnitude make data collection extremely difficult and fully parameterized models like we are familiar with for routine grid operation, are not feasible. We feel that given the complexity and challenges in obtaining the necessary measurements for this problem, the approach we have taken is the best possible. We look forward to future work to elucidate the details of the mechanisms that dominate in different scales of infrastructure damage and recovery. However, as the reviewer mentioned in a previous comment, adding anything more to our model would almost certainly make it "unnecessarily complicated" in light of the quality of data that is accessible at this stage.

Yet, the data is sufficient to show a new direction for data collection and for network modeling, bringing under the same roof the previously unconnected findings in disaster literature and network science. To wait until we have a complete theory for all scales of damage would do a disservice to the researchers and stakeholders who need tools and data to cope with disasters.

The case study of Imelda is important for us because it gave us an opportunity to explicitly compare a hypothesis of resource constraints and recovery coupling as mechanisms, and select recovery coupling as the operative mechanism. We hope that with the publication of this paper, the associated data, and the clear case of Imelda, it will inspire others to discover further case studies that capture all potential mechanisms.

Regarding the suggestion of a threshold model for repair rate slowdown, we would like to draw the reviewer's attention to equations 1 and 2 of the SI. Equation 1 defines a self-consistent relationship between γ^r and f --it is linear in f but includes a $\gamma^r * f$ term, making it non-linear overall. As predicted by the reviewer, this in fact leads to the $1/f$ behavior which the reviewer has hypothesized should

occur. This leads to results that are essentially the same as the strict threshold model, with repair rate saturating as damage increases, but with continuously differentiable functions. And--like that model--would be consistent with most of the observations.

(C) More generally, I think that the framework of interdependent networks is overrated, if not misleading, in the context of power grid stability and reliability. The highly cited reference [3] has established interdependent networks using the example of the 2003 Italian blackout. From an engineering viewpoint, this is highly problematic. Communication networks did not play a role in this cascade, while elementary aspects of power engineering did, but were not included in the analysis. The authors of the current paper discuss this fact on page 7, which is good.

Still, the authors proceed in a similar way as in reference [3]. The first part of the manuscript is about empirical data and the second part is about a highly abstract model of interdependent networks. In my opinion, both parts are only very weakly connected. As argued above the interdependent network approach is not needed to analyze the data, in the simplified version analyzed in Fig. 4 it is certainly not adequate to describe power system reliability. From an engineering perspective, the figure 4 and the associated discussion weaken the paper.

We fully agree with the reviewer's sentiment that the models based on domino-like dependency have much more limited applicability than has been widely appreciated. Indeed, we launched this line of inquiry when we discovered the lack of clear-cut examples of domino-like dependency, even as there were many anecdotal examples of recovery or restoration interdependencies. The fact that the former case had been modelled in countless variations while lacking empirical data to support it, while the empirically observed dependency has been largely overlooked is a major motivation for our study. As we mentioned in previous comments, we are not claiming that our model is the final word on power systems reliability. All we are claiming is that the concept can be coherently captured mathematically, that measurements can be made which fit the predictions of the model, and that there are cases like Imelda, in which we can show with high confidence that the mechanism of recovery coupling is at play. We maintain that these are substantial improvements on the state of the art, hence that our results could inspire further work to better understand power systems reliability in its broader socio-technological context.

(D) Finally, one should be aware that the majority of power outages (the smaller ones) does not involve any cascading mechanism. Most outages result from equipment failures in distribution grids or feeders at low or middle voltage grids, which are not or only weakly meshed. Then the loss of a single line can directly disconnects some consumers. An analysis of such outages is often possible without a detailed network approach.

On the contrary, high voltage transmission grids are meshed such that the loss of a single line should not lead to any disconnected elements if the grid is operated properly (this is called the N-1 criterion). However, in some cases we nevertheless see cascading failures leading to large scale outages disconnecting thousand or millions of consumers. Understanding these cascades does require a detailed network approach including the physics of power flows.

This is an excellent point, and one of the main reasons that we focused on percolation-based models and not flow-based cascade models in our analysis. The percolation models capture the fact mentioned by the referee that things get cut off, without invoking complex assumptions about power load balancing. We fully agree with the reviewer that this is indeed the explanation for the vast majority of the outages. We also agree with the reviewer that this point is widely overlooked in the literature. In fact, we are currently pursuing a new research project which aims to better characterize the distinct signatures of distribution vs transmission outages based on empirical data. We hope that by following the reviewer's suggestion and releasing the data along with the paper, we can encourage wider appreciation of this fact.

Minor issues:

(1) How is „recovery coupling“ defined exactly? The term is used a lot, but a precise definition is lacking. Figure 1 gives an idea (which is nice as such), but not a definition.

Thank you for drawing our attention to this point. We define recovery coupling as the dependence of the recovery of one infrastructure system on the functional state of another infrastructure system. We have added this to the abstract following the reviewer's suggestion.

(2) Figure 2 gives a compelling evidence that power grid recovery depends on accessibility via road transports. But is this an evidence for “recovery coupling”? The paper and in particular figure 1 imply that recovery coupling is only defined for network but the network aspects are not really present in figure 3. This is not a criticism of the figure as such (which I find great), I just think that the caption is way to unspecific.

We thank the reviewer for pointing this out, and we have adjusted the caption to make it more specific. Because the road network is also an infrastructure system, the impact of road activity/access on the recovery of power or other infrastructure is one of the many possible examples of recovery coupling. For the purposes of this analysis, we have taken proximity to damage as a proxy for the network effects of a road closure. A more accurate estimate could be obtained by including explicit traffic modeling, which would be an excellent direction for future research, but is beyond the scope of this work.

(3) As mentioned in the beginning, an open outage repository would be of extraordinary value for power system reliability science. However, the description in the main manuscript is not precise enough. For instance, what exactly do you mean if you count an “outage”? Is it one event where a certain number of customers is disconnected? Or does one outage correspond to exactly one failing element of the grid?

(4) Extending on my previous comment: A methods section in the main manuscript would clearly be very helpful to understand what is done exactly.

We apologize for the lack of clarity. We will be releasing all of the data that we used in the analysis, to serve as a resource for future researchers, as recommended by the referee. It is available at

<https://github.com/mmdanziger/recovery-coupling>

In the new methods section we have detailed exactly what an "outage" means. We inherit the definition of an outage from the utility's outage map. From observation and discussions with local utility representatives

we learned that a reported outage represents a single failing element, according to the utility's understanding of the situation. To determine the duration of an outage, we link observed outages from snapshot to snapshot, using the location, start time and reported unique id. We describe this in the new methods section.

(5) Page 4. The authors write “all large events collapse on a single nonlinear curve.” I am not fully convinced by this statement. There is still quite a large degree of variability in the data. Furthermore, when I look into the curves in the Supplementary Figures 2 and 3, I see quite strong differences.

We thank the referee for bringing this to our attention, we have rephrased the statement. It now reads "all large events display similar nonlinear behavior".

(6) On page 7, the authors summarize their results and give an outlook in the paragraph starting with “The ability to identify”. In my opinion, the conclusions are a little exaggerated. Does this single paper “open the door” to the large field of infrastructure vulnerability? I recommend to tone down the voice here.

We apologize for giving the wrong impression with regard to previous work. We wanted to emphasize that within the field of interdependent network resilience, there has been a near total lack of predictions that can be directly compared with empirical measurements of infrastructure damage. By shifting the focus to the recovery process, we are able to model dynamic interdependence in a way that can be validated directly with empirical measurements. Such a data-driven infrastructure interdependence case study has never been successfully accomplished, to our knowledge.

We expect that in time, many more examples of recovery coupling will be measured, as well as examples of simpler mechanisms such as the resource constraints that the reviewer suggested, and that we discussed in the SI. We have shown how the question of what mechanism is at play can be posed and--at least sometimes--answered. We hope that our paper will inspire other researchers to ask the very questions that the referee has raised, namely what mechanism is at play in a given situation.

In light of the reviewer's comment we have rephrased the paragraph. It now reads:

The data-driven approach presented here enables a more precise understanding of infrastructure interdependence. For example, we find that while the set of flooded roads as a whole caused slowdowns in power outage repairs, some impaired roads had much stronger effects than others. The roads in downtown Houston caused only minor delays when flooded, while in Beaumont and Northeast Houston flooded roads caused severe delays (see Fig. 3a). Improving the precision with which we measure infrastructure vulnerability is particularly important in light of aging infrastructure and climate change.

(7) Finally I am a bit surprised that a study on power system reliability contains so few references to papers in power engineering journals. For instance, there are ongoing discussions about self-healing smart power grids – a topic which could be of relevance here.

A possible starting point in this direction can be: Amin, Massoud. "Challenges in reliability, security, efficiency, and resilience of energy infrastructure: Toward smart self-healing electric power grid." 2008 IEEE Power and energy society general meeting-conversion and delivery of electrical energy in the 21st century. IEEE, 2008.

A discussion of the role of storms on distribution grid reliability may be found here:

Brown, R. E., et al. "Distribution system reliability assessment: momentary interruptions and storms." IEEE Transactions on power Delivery 12.4 (1997): 1569-1575.

Some statistical studies of distribution grids reliability indices may be found here:

Balijepalli, Nagaraj, Subrahmanyam S. Venkata, and Richard D. Christie. "Modeling and analysis of distribution reliability indices." IEEE Transactions on Power Delivery 19.4 (2004): 1950-1955.

Finally, I strongly recommend to read this great review article on cascading failures and power system blackouts:

Pourbeik, Pouyan, Prabha S. Kundur, and Carson W. Taylor. "The anatomy of a power grid blackout-Root causes and dynamics of recent major blackouts." IEEE Power and Energy Magazine 4.5 (2006): 22-29.

We thank the reviewer for sharing these papers, and we have added references to Brown 1997 and Poureik 2006 which we hope will provide our readers with better context regarding previous work in the field.

Reviewer #3 (Remarks to the Author):

The paper introduces a theoretical framework based on nonlinear systems and multi-layer networks, with an application on power networks.

The paper is clearly written and presents a convincing framework elegant in its simplicity, that is able to capture the main aspects related to the interdependence of power infrastructures in relation to telecommunications and road networks.

The database considered in the study is consistent and covers a wide temporal and geographical frame.

Results are presented in a clear and convincing way, and in particular the existence of two stable solutions is of paramount importance for both utilities and decision makers that aim to develop tools for the analysis and prediction of catastrophic events, even those not directly related to storms or other atmospheric disasters, as, by example, climate-related issues like heat waves and increased systems failure due to temperature increase or decrease.

We wish to thank the referee for the succinct summary of our results and the positive assessment of the manuscript.

In my opinion the paper is certainly worth to be published after the following minor points:

1) Please provide a better description about the events that you did consider for the development of the model. Did the authors only consider big failure events or they also included in the analysis other events that can be easily fixed by automatic procedures? (e.g. disconnection of overloaded stations, failure of TLC systems etc).

Thank you for this suggestion. We have included a more detailed description of the data used in the new methods section. We are also releasing the data to the public at

<https://github.com/mmdanziger/recovery-coupling>

2) Please include in the conclusions implications for industry, utilities and policy/decision makers.

We have added a new paragraph to the conclusion where we discuss the implications of our model for policy-makers and utilities. The most significant takeaway is that in order to maintain robust infrastructure, it is critical not only to strengthen a single infrastructure system, but also to introduce measures to enhance resilience of the other infrastructure systems needed to restore it.

REVIEWERS' COMMENTS

Reviewer #1 (Remarks to the Author):

The Authors have addressed all my comments. I now judge this work suitable for publication in Nature Communication.

Reviewer #2 (Remarks to the Author):

Report on the manuscript Recovery Coupling in Multilayer Networks by Danziger and Barabasi.

The authors have revised their manuscript based on the comments of the three reviewers.

I think that the revision has substantially improved the paper. The compiled data set is very useful, the quantitative analysis is thorough and the insights are valuable. My personal view of the paper is still ambivalent as the emphasis on interdependent networks appears somewhat artificial from an engineering viewpoint. However, this is my personal view and should not prevent the results from being published.

I have two final comments before I will recommend publication:

(1) Please cite the excellent paper "Reducing cascading failure risk by increasing infrastructure network interdependence" by Korkali, Veneman, Tivnan, Bagrow, and Hines (Scientific reports, 2017) in the discussion. This paper shows most clearly that interdependencies can have quite diverse impacts on power grid stability.

(2) Please think about the title of the paper. It is highly unspecific and does not give a very good impression about the actual content of the paper.

Reviewer #3 (Remarks to the Author):

I believe the paper improved from the first version, I strongly support publication for this paper

Recovery Coupling in Multilayer Networks

Point-by-point response to reviewer comments

Reviewer #1 (Remarks to the Author):

The Authors have addressed all my comments. I now judge this work suitable for publication in Nature Communication.

We thank the reviewer for their time and support.

Reviewer #2 (Remarks to the Author):

Report on the manuscript Recovery Coupling in Multilayer Networks by Danziger and Barabasi.

The authors have revised their manuscript based on the comments of the three reviewers.

I think that the revision has substantially improved the paper. The compiled data set is very useful, the quantitative analysis is thorough and the insights are valuable. My personal view of the paper is still ambivalent as the emphasis on interdependent networks appears somewhat artificial from an engineering viewpoint. However, this is my personal view and should not prevent the results from being published.

We are gratified that our revision has addressed the issues that concerned the reviewer, and we appreciate their support for publishing the paper.

I have two final comments before I will recommend publication:

(1) Please cite the excellent paper "Reducing cascading failure risk by increasing infrastructure network interdependence" by Korkali, Veneman, Tivnan, Bagrow, and Hines (Scientific reports, 2017) in the discussion. This paper shows most clearly that interdependencies can have quite diverse impacts on power grid stability.

This is indeed an interesting reference and we have included it in the discussion section.

(2) Please think about the title of the paper. It is highly unspecific and does not give a very good impression about the actual content of the paper.

We acknowledge that the current title may not communicate the full content of the research, but upon consideration we believe it to be the best option.

Reviewer #3 (Remarks to the Author):

I believe the paper improved from the first version, I strongly support publication for this paper

We thank the reviewer for their time and support.